# Pro-Fibrotic Macrophage Subtypes: SPP1+ Macrophages as a Key Player and Therapeutic Target in Cardiac Fibrosis?

**DOI:** 10.3390/cells14050345

**Published:** 2025-02-27

**Authors:** Moritz Uhlig, Sebastian Billig, Jan Wienhold, David Schumacher

**Affiliations:** 1Department of Anesthesiology, Faculty of Medicine, RWTH Aachen University, 52074 Aachen, Germany; 2Department of Medicine 2 (Nephrology, Rheumatology, Clinical Immunology and Hypertension), Faculty of Medicine, RWTH Aachen University, 52074 Aachen, Germany

**Keywords:** fibrosis, macrophages, cardiac fibrosis, organ fibrosis, Spp1, Trem2, ECM remodeling, macrophage subsets, fibrotic niche

## Abstract

Cardiac fibrosis is a major driver of heart failure, a leading cause of morbidity and mortality worldwide. Advances in single-cell transcriptomics have revealed the pivotal role of SPP1+ macrophages in the pathogenesis of cardiac fibrosis, positioning them as critical mediators and promising therapeutic targets. SPP1+ macrophages, characterized by elevated expression of *secreted phosphoprotein 1* (*SPP1*) and often co-expressing *Triggering Receptor Expressed on Myeloid Cells 2* (*TREM2*), localize to fibrotic niches in the heart and other organs. These cells interact with activated fibroblasts and myofibroblasts, driving extracellular matrix remodeling and fibrosis progression. Their differentiation is orchestrated by signals such as CXCL4, GM-CSF, and IL-17A, further emphasizing their regulatory complexity. Therapeutic strategies targeting SPP1+ macrophages have shown encouraging preclinical results. Approaches include silencing *Spp1* using antibody–siRNA conjugates and modulating key pathways involved in macrophage differentiation. These interventions have effectively reduced fibrosis and improved cardiac function in animal models. The mechanisms underlying SPP1+ macrophage function in cardiac fibrosis provide a foundation for innovative therapies aimed at mitigating pathological remodeling and improving outcomes in patients with heart failure. This emerging field has significant potential to transform the treatment of fibrotic heart disease.

## 1. Introduction

With more than 64 million cases worldwide, chronic heart failure is one of the most common in-hospital diagnoses [1]. It places a dramatic burden on patients, as it leads to a marked reduction in their fitness and a limited capacity to manage their everyday life. Patients with severe heart failure ultimately suffer from cardiac failure due to a lack of therapeutic options with a five-year mortality of 50% for men and 46% for women [2,3].

Given the limited efficacy of previous strategies in developing new therapies for heart failure, cardiac fibrosis has emerged as a key area of current research, revealing promising therapeutic targets [4].

Currently available therapeutic strategies delay the progression of heart failure itself but have a very limited effect on adverse cardiac remodeling. To date, strategies targeting the broadly implicated inflammatory and pro-fibrotic pathways have failed to demonstrate effectiveness in clinical practice [5].

Macrophages play a pivotal role in maintaining tissue homeostasis by phagocytosing, degrading, and removing cellular debris and dead cells [6]. In addition, they have a reparative function and play an essential role in tissue repair, e.g., after myocardial infarction (MI) [7]. However, they are not directly responsible for the deposition of fibrosis-typical extracellular matrix (ECM) products [8,9,10,11,12,13] but are considered to indirectly control fibrogenesis through the stimulation of ECM-secreting mesenchymal cells and modulation of the ECM [14]. Whereas macrophage-modulated ECM production is essential for tissue repair (e.g., replacement fibrosis for scar formation after MI to avoid dilatation and cardiac rupture), dysregulated and maladaptive ECM production leads to organ fibrosis, ultimately leading to organ failure due to a disrupted architecture. Consequently, macrophages are considered a promising target for the treatment of organ fibrosis. Indeed, the modulation of macrophage states might reduce ECM production. Recent studies leveraged transcriptomic data of mononuclear cells to identify new targets for the treatment of cardiac fibrosis and heart failure [15,16]. Some of these studies revealed *secreted phosphoprotein 1*-positive (SPP1+) macrophages as a promising target, and there has been a growing interest since then [17,18,19].

This brief review aims to synthesize the current state of research regarding the role of pro-fibrotic macrophage subtypes in organ fibrosis, with a particular emphasis on cardiac fibrosis and the specific contributions of SPP1+ macrophages.

The study of pro-fibrotic macrophages in cardiac fibrosis necessitates an integrative understanding of general organ fibrosis due to conserved molecular pathways and cellular interactions across different tissues. Fibrosis in organs such as the liver, lungs, and kidneys shares several core mechanisms with cardiac fibrosis, including common signaling pathways and macrophage plasticity. Due to these parallels between fibrotic mechanisms in different organs, the discussion will begin with a comprehensive overview of the cellular and molecular mechanisms underlying organ fibrosis. It will highlight the pivotal roles of various macrophage subtypes and their functional characteristics. Insights from fibrotic processes in other organs provide valuable mechanistic templates for understanding cardiac fibrosis.

Special attention will be given to SPP1, which has emerged as a key marker and mediator in pro-fibrotic pathways, particularly in the context of pathological cardiac remodeling. In addition, it will explore potential therapeutic strategies aimed at modulating these macrophage populations to mitigate fibrotic progression and improve clinical outcomes.

## 2. Fibrosis

Myocardial fibrosis arises from various triggers. Myocardial infarction causes replacement fibrosis due to cardiomyocyte necrosis, whereas pressure or volume overload conditions like hypertension and valve regurgitation rather induce reactive fibrosis through mechanical stress and neurohormonal activation [20,21]. Genetic disorders such as Fabry disease promote infiltrative fibrosis via glycosphingolipid accumulation, while metabolic conditions like diabetes and obesity exacerbate fibrosis through chronic inflammation and oxidative stress [22,23].

Fibrosis is not a disease per se, but rather a provisional response of the tissue to various forms of injury. It results of the subsequent physiological repair reaction in which there is an increasing deposition of ECM components (including collagens and fibronectin). However, this can take on a pathological value if dysregulation occurs due to repetitive trauma or chronic inflammatory processes [24].

In response to tissue trauma, there is a local activation of fibroblasts, which subsequently increases their proliferation and migration and upregulates the secretion of inflammatory mediators and the synthesis of ECM components. Moreover, fibroblasts differentiate into myofibroblasts, which produce even more ECM and possess contractile properties to pull together the injured tissue (e.g., this reduces the risk of rupture after myocardial infarction).

In the case of minor or singular injuries, wound healing is typically efficient, with the accumulation of ECM components being transient and subsequently broken down at a rapid pace. This results in a restitutio ad integrum. However, repeated or severe tissue trauma can result in the excessive deposition of ECM components, which can consecutively disrupt the tissue architecture. This can result in the dysfunction of the affected organ, which may ultimately lead to organ failure. Consequently, fibrosis is deemed to be a significant contributor to mortality in the industrialized world, with an estimated 45% of all deaths attributed to this condition [24].

As mentioned above, myocardial fibrosis manifests in two different patterns. Replacement fibrosis occurs after irreversible injury, such as myocardial infarction, in which necrotic cardiomyocytes are replaced by a dense, collagen-rich scar that, although critical for maintaining structural integrity, irreversibly impairs contractile function [25]. In contrast, reactive (interstitial or perivascular) fibrosis is characterized by a more diffuse accumulation of extracellular matrix proteins in response to chronic stressors (e.g., pressure overload or prolonged inflammatory stimuli) without necessarily involving apparent cell death [25].

Such disruption of tissue architecture leads to significant clinical complications. The replacement of healthy cardiac tissue with non-functional fibrous tissue impairs myocardial elasticity, compromising both systolic and diastolic functions. Clinical manifestations are dyspnea, chronic fatigue, and edema [26,27]. Additionally, fibrotic tissue disrupts normal electrical conduction, leading to arrhythmias such as atrial fibrillation and ventricular tachycardia, which may result in syncope or sudden cardiac death [26,28]. In advanced stages, symptoms like angina pectoris and dizziness may occur even at rest due to inadequate myocardial oxygenation [29]. Furthermore, extensive fibrosis is associated with an increased risk of complications including higher mortality rates following interventional procedures and a heightened susceptibility to myocardial infarction [30]. In contrast to replacement fibrosis, reactive fibrosis may be reversible if the underlying cause is eliminated.

## 3. SPP1

Osteopontin (SPP1) is a matricellular protein encoded by the SPP1 gene that promotes the recruitment of monocyte-macrophages and regulates cytokine production in macrophages, dendritic cells, and T cells, among others. In its capacity as a T_H_1 cytokine, it is thought to increase inflammation in several chronic inflammatory diseases, including atherosclerosis [31,32]. In contrast to structural matrix proteins, matricellular proteins do not play a primary role in tissue architecture but are only induced after tissue injury and modulate cell–cell and cell–matrix interactions. Osteopontin activates inflammatory cells and promotes cell survival, adhesion, and migration [25].

## 4. SPP1+ Macrophages in Fibrosis

After tissue injury, myeloid cells are recruited from the bloodstream and follow a chemokine ligand 2 (CCL2) gradient to the injured tissue where they differentiate into macrophages, a key cell population in the removal of apoptotic cells and tissue debris alongside the degradation and remodeling of the ECM, as already mentioned above [33]. This removal is crucial to allow proper tissue repair. Triggered by phagocytosis of ECM fragments such as hyaluronan, proinflammatory macrophages secrete chemokines and proinflammatory cytokines as well as several cytotoxic mediators that may contribute to the progression of fibrosis if macrophage activity is dysregulated [34,35].

It has long been thought that there are two different subtypes/activation states of macrophages: inflammatory macrophages (M1) and reparative macrophages (M2). Depending on the activation pathway, macrophages can either initiate and maintain chronic inflammation (M1, classical activation) or secrete pro-fibrotic factors and aid tissue repair (M2, alternative activation) [36].

However, more and more studies suggest that this dichotomous classification is outdated. High-resolution transcriptomic and single-cell analyses have revealed that macrophages exist in a highly dynamic and context-dependent spectrum of activation states rather than distinct M1 or M2 subsets. At least from a transcriptional point of view, several subtypes were identified, some of them even expressing inflammatory and reparative genes, suggesting that there might be a transition between cell states. Thus M1/M2 dichotomy oversimplifies macrophage plasticity and fails to account for specialized, disease-associated macrophage populations such as lipid-associated macrophages (LAMs) [37], disease-associated macrophages (DAMs) [38], tumor-associated macrophages (TAMs) [39], and scar-associated macrophages (SAMs) [40,41,42]. It was demonstrated that macrophages exhibit complex transcriptional signatures that adapt to microenvironmental cues, emphasizing the need for a more refined, functional, and disease-specific classification beyond the M1/M2 paradigm.

In a recent study, Coulis et al. utilized single-cell transcriptomics to investigate the molecular characteristics of dystrophic and healthy muscle macrophages. This approach enabled the identification of six novel clusters that did not align with existing definitions of M1 or M2 macrophages. Among them, two distinct subsets were identified as exhibiting pro-fibrotic properties: monocyte-derived macrophages (MDMs) and Galectin3+ (Gal3+) macrophages. MDMs were characterized by the high expression of markers like *Cd52*, *Plac8*, *Prdx5*, and *Hp* and were elevated in early stages of muscular dystrophy but resolved by chronic stages of the disease, whereas the predominant macrophage signature in dystrophic muscle was characterized by high expression of fibrotic factors *Lgals3* and *Spp1* (cf. Table 1, Figure 1) [43]. These Gal-3+ macrophages are elevated in dystrophic muscles and exhibit a specific transcriptional profile associated with ECM remodeling and fibrotic processes. Spatial transcriptomics revealed that Gal-3+ macrophages are localized in regions of high fibrosis and interact closely with stromal cells. It also showed an upregulation of ECM genes (*Fn1*, *Postn*, *collagens*) and fibrosis-inducing growth factor pathways (TGFβ, PDGF) in Gal-3-enriched areas (cf. Figure 2A). In vitro analyses demonstrated that these macrophages regulate the differentiation of stromal progenitor cells through SPP1. Furthermore, adoptive transfer experiments demonstrated that MDMs in dystrophic muscle upregulated genes associated with Gal-3+ macrophages, suggesting they can transition to this phenotype in the dystrophic environment. Additionally, Gal-3+ macrophages are elevated in several human myopathies, indicating a conserved role for these cells in fibrosis. They represent a specialized population that plays a crucial role in the onset and progression of skeletal muscle fibrosis.

In line with that, a macrophage subtype characterized by high expression of *CD9* and *TREM2* was identified, referred to as “scar-associated macrophages” (SAMs or SAMacs), which were further specified by single-cell RNA sequencing (scRNA-seq) data from human and murine fibrotic tissues (cf. Table 1, Figure 1) [40,41]. It has been demonstrated that the additional expression of *SPP1* is indicative of a pro-fibrotic phenotype.

Ramachandran et al. first described SAMacs in the context of liver fibrosis. Their findings suggest that, from a functional perspective, SAMacs originate from circulating monocytes and display a hybrid phenotype, incorporating characteristics of both tissue monocytes and Kupffer cells. This SAMac subpopulation could be identified by the additional expression of *Lgals3* and *Spp1* (*Cd9*, *Trem2*, *Spp1,* and *Lgals3*) in mice and *SPP1, TREM2,* and *CD9* in humans [41]. They accumulate in the fibrotic niche through local proliferation and appear to play a role in ECM remodeling, tissue fibrosis, and angiogenesis (cf. Table 1, Figure 1).

SAMacs have been found to express profibrogenic genes including *SPP1*, *LGALS3*, *CCL2*, *CXCL8*, *PDGFB*, and *VEGFA* and are strategically located in collagen-rich scar regions of cirrhotic livers. They secrete key mediators of hepatic fibrosis promoting fibrillar collagen expression and proliferation in hepatic stellate cells (HSCs). Furthermore, SAMacs express EGF receptor (EGFR) ligands and mesenchymal cell mitogens such as tumor necrosis factor (ligand) superfamily member 12 (*TNFSF12*) and platelet-derived growth factor B (*PDGFB*), which interact with receptors on scar-associated mesenchymal cells, thereby stimulating their activation and proliferation (cf. Figure 2A) [41].

Fabre et al. were able to detect SAMs in both liver and the lung tissue [40]. These SAMs, which were most specifically described by the expression of *SPP1*, *CD9*, *TREM2*, *GPNMB,* and *FABP5*, were located at the periphery of scar tissues, in proximity to activated mesenchymal cells, and were also associated with a fibrotic niche (cf. Table 1, Figure 1).

In an earlier study, Fabre et al. already introduced the term “type 3 inflammation” by pointing out common mechanisms with type 3 immunity [46]. Both concepts are driven by type 3 cytokines (granulocyte-macrophage colony-stimulating factor (GM-CSF), interleukin 17A (IL-17A), and interleukin 22 (IL-22)) produced by neutrophils. However, while type 3 immunity is primarily directed against pathogens, type 3 inflammation leads to chronic activation of macrophages (Fab5 SAMs) and fibroblasts that induce TGF-β-dependent fibrosis. The common role of type 3 cytokines in immune activation and tissue remodeling provides the rationale for this term.

Neutrophils that produce the type-3 cytokines GM-CSF and IL-17A, as well as matrix metalloproteinase 9 (MMP9), which is involved in the activation of transforming growth factor beta 1 (TGF-β1), are present alongside these SAMs. In vitro studies indicate that GM-CSF, IL-17A, and TGF-β1 drive the differentiation of these SAMs. Therapeutic blockade of GM-CSF, IL-17A, or TGF-β1 reduces the expansion of SAMs and fibrosis in liver and lung fibrosis models. The presence of SAMs has been observed to promote fibrosis by maintaining a pathological ECM turnover and type 3 inflammation. The pro-fibrotic signaling of TGF-β and SMAD3 has been shown to result in increased fibroblast and myofibroblast activation, effecting the degradation of regular ECM components and the deposition of collagen I. This pathological turnover is associated with the activation of MMPs, myofibroblasts, and T_H_17 cells (Figure 2A) [40].

Both studies provide valuable insights into the role of SAMs in fibrosis [40]. However, Fabre et al. offer a more comprehensive understanding of their interaction with neutrophils and cytokine-driven differentiation, while Ramachandran et al. provide detailed insights into their transcriptional profile and origins.

ScRNA-seq has identified three distinct subsets of pulmonary macrophages in both normal and idiopathic pulmonary fibrosis (IPF) lungs: FABP4^hi^ macrophages (characterized by high expression of Fatty Acid Binding Protein 4 (*FABP4*) and Inhibin Subunit Beta A (*INHBA*) but low *SPP1* and MER Tyrosine Kinase (*MERTK*)), FCN1^hi^ macrophages (with high Ficolin-1 (*FCN1*) expression), and SPP1^hi^ macrophages (high expression levels of *SPP1* and *MERTK*, cf. Table 1, Figure 1) [45]. In IPF lungs, the lower lobes showed a marked increase in SPP1^hi^ macrophages and significant SPP1 deposition. The upper lobes had fewer SPP1^hi^ macrophages and more FABP4^hi^ macrophages, similar to normal lungs [45].

Proliferation of SPP1^hi^ macrophages was low in normal lungs but dramatically increased in IPF lungs. Both FABP4hi and SPP1^hi^ macrophages exhibited higher proliferation rates in IPF, indicating their active role in disease progression. Co-localization and causal modeling revealed that SPP1^hi^ macrophages are crucial for activating myofibroblasts, significantly contributing to fibrosis in IPF [45]. SPP1^hi^ macrophages show highly upregulated expression of *MERTK*. As a receptor for apoptotic cells, MERTK enables these macrophages to clear apoptotic cells and promote tissue repair. The efferocytosis mediated by MERTK suppresses inflammation and contributes to tissue repair through the upregulation of TGF-β, prostaglandin E2 (PGE2), and platelet-activating factor (PAF) [47]. It has been suggested that the increased co-expression of *SPP1* and *MERTK* in these macrophages may play a pivotal role in their function in tissue repair and fibrosis in IPF (cf. Figure 2A). This co-expression may contribute to the activation of myofibroblasts and the progression of lung fibrosis.

It can be reasonably assumed that SPP1+ macrophages represent a heterogeneous cell population (cf. Table 1, Figure 1). According to this hypothesis, Ouyang et al. describe a polarization state of SPP1+ macrophages, termed matrisome-associated macrophages (MAMs), which have been recognized in the liver and lungs [42]. The authors distinguish between SPP1+ MAM− and SPP1+ MAM+ macrophages, with MAM− essentially representing the SPP1+ macrophages that do not exhibit marker genes indicative of MAM+ polarization. Through the integration of 15 scRNA-seq datasets, the characteristics of SPP1+ MAM+ macrophages have been further elucidated, revealing their presence in the fibrotic tissues of the heart, lung, liver, skin, endometrium, and kidney. This cell population is markedly elevated in fibrotic tissues, suggesting a potential role in the progression of cardiac as well as multiorgan fibrosis (cf. Figure 2B) [44]. While the authors provide valuable insights into SPP1+ macrophage heterogeneity, its biological relevance must be critically assessed. Without robust validation, the functional significance of identified subclusters remains speculative. Careful integration with complementary data, such as spatial transcriptomics or functional assays, is essential to ensure that sub clustering results accurately reflect true biological phenomena rather than computational overfitting.

In accordance with the findings from other organ systems, the presence of SPP1+ macrophages has also been observed in the context of cardiac fibrosis. A population of macrophages with high expression of *Spp1*, *Fibronectin 1* (*Fn1*), and *Arginase 1* (*Arg1*) has been described in mice by Hoeft et al. (*Spp1+* macrophages), which expands following organ injury (cf. Table 1, Figure 1) with a corresponding SPP1+ macrophage subset found in human tissue (*SPP1*, *TREM2*, *CD9*, *FN1*, *APOE*) [18]. CXCL4, also known as platelet factor 4 (PF4), has been identified as an essential chemokine for the differentiation of these pro-fibrotic macrophages (cf. Figure 2B). Both in vitro and in vivo studies demonstrated that the loss of CXCL4 prevents the differentiation of *Spp1+* macrophages and reduces fibrosis after heart and kidney injuries. Platelets, the primary source of CXCL4 in the body, drive the differentiation of pro-fibrotic *Spp1+* macrophages. Single-nucleus RNA sequencing (snRNA-seq) and ligand–receptor interaction analysis revealed that macrophages mediate the activation of myofibroblasts through *Spp1*, *Fn1*, and *Semaphorin 3* (*Sema3*). Additionally, *SPP1+* macrophages expand in both chronic kidney disease and heart failure in humans. Therefore platelet-instructed *SPP1+* macrophages play a crucial role in the activation of myofibroblasts and the promotion of fibrosis through a CXCL4-dependent mechanism [18]. Blocking CXCL4 could represent a potential therapeutic strategy to reduce fibrosis across various organs.

Kuppe et al. assessed the role of SPP1+ macrophages in the heart in greater detail by providing a comprehensive and high-resolution map of human cardiac tissue remodeling following myocardial infarction (MI) [17]. By integrating scRNA-seq data, chromatin accessibility data (ATAC-seq), and spatially resolved transcriptomic data from various physiological zones and time points of the myocardium in patients with myocardial infarction and controls, they identified SPP1+ macrophages (*SPP1*+ *CD36*+) as a significant cell type that expands in ischemic heart tissues following myocardial infarction (MI; cf. Table 1, Figure 1). This macrophage population plays a crucial role in remodeling the ECM and promoting fibrotic processes in the heart. Molecular characterization of SPP1+ macrophages reveals their high expression of genes associated with ECM remodeling and inflammatory signaling pathways. Between SPP1^+^ macrophages and myofibroblasts, particularly in ischemic samples, an enhanced PDGF-C, PDGF-D, and Thrombospondin-1 (THBS1) signaling was observed (cf. Figure 2B). SPP1+ macrophages were also enriched in areas of intense fibrosis and are found near activated fibroblasts, underscoring their role in the fibrotic response [17].

Moreover, it was seen that CCL18+ macrophages were increased in fibrotic samples. Spatial analysis revealed that they closely interacted with SPP1+ macrophages, with SPP1+ macrophages predicting the presence of CCL18+ macrophages. Additionally, the authors observed distinct changes in cellular crosstalk between SPP1+ macrophages and fibroblasts in fibrotic samples, including increased ADAM17 and TGFB1 signaling compared to myogenic samples (cf. Figure 2B) [17].

Spatial transcriptomics has demonstrated that SPP1+ macrophages are specifically localized in fibrotic niches. This spatial arrangement indicates that SPP1+ macrophages are closely associated with the pathogenesis of fibrosis and tissue remodeling post-MI. Moreover, the presence of SPP1+ macrophages within fibrotic niches emerged as the subtype with the highest predictive probability for activated myofibroblasts, supporting the hypothesis that SPP1+ macrophages exacerbate fibrosis through their interaction with fibroblasts. Additionally, SPP1+ macrophages interact with other immune cells, contributing to the maintenance of a pro-fibrotic microenvironment.

The identification of SPP1+ macrophages as key players in the fibrotic remodeling of the heart post-MI offers potential therapeutic targets. Understanding the mechanisms that regulate the activity of these macrophages could lead to the development of new strategies for treating cardiac fibrosis.

## 5. TREM2

Another frequently reported marker of pro-fibrotic macrophage populations is *TREM2*. These TREM2+ macrophages may express *SPP1* and are involved in liver and lung fibrosis, as well as fibrosis after myocardial infarction and atrial fibrillation [18,19,40]. They interact extensively with the ECM and other cell types such as myofibroblasts to support tissue remodeling and promote fibrosis.

In the context of cardiac fibrosis associated with heart failure, SPP1+ macrophages expressing *TREM2* and *CD9* showed an increased presence and were strongly correlated with fibrotic changes in the heart tissue [17,18,40,44]. The co-expression of *TREM2* and *SPP1* therefore appears to be a relevant factor. In line with this, *Hoeft* et al. have identified TREM2+ SPP1+ macrophages as a specialized population that plays an important role in the development and progression of fibrosis [18].

Not all TREM2 macrophages express *SPP1*. The co-expression of *TREM2* and *SPP1* is characteristic of a specific subset of macrophages known as lipid-associated macrophages (LAMs). This subset has been particularly identified in areas of tissue damage and fibrosis [48]. However, there are other populations of TREM2-expressing macrophages that do not necessarily express *SPP1*. Some studies have demonstrated that TREM2+ macrophages are present in different disease models and tissues and can perform diverse functions depending on the specific signals and microenvironments to which they are exposed. This indicates the complexity and versatility of TREM2+ macrophages in response to varying pathological contexts [49,50].

## 6. Limitations of Sequencing-Based Studies

Despite the significant insights provided by high-throughput sequencing techniques in the studies analyzed, several limitations must be considered when interpreting their biological relevance.

A significant limitation is the reliance on transcriptional data alone, as gene expression at the RNA level does not always correlate directly with protein abundance and function. While some studies incorporated proteomic validation methods such as immunofluorescence, flow cytometry, or Luminex assays, others primarily relied on transcriptomic and computational approaches without direct proteomic validation. This raises concerns regarding the extent to which observed transcriptomic changes translate into functional protein-level alterations, which are essential for understanding cellular behavior and disease mechanisms. Transcriptomic data should always be validated to prove biological relevance.

Furthermore, the lack of functional genetic perturbations (e.g., gene knockdowns or knockouts) in some studies limits the ability to establish causal relationships between identified gene expression patterns and biological processes. While Hoeft et al. implemented a CXCL4 knockout model to investigate fibrosis-related mechanisms, other studies primarily inferred biological roles from correlation-based transcriptomic analyses rather than direct functional validation. This absence of mechanistic interrogation may lead to overinterpretation of gene regulatory networks and their impact on disease progression.

Some studies, such as Kuppe et al. [16], have already implemented multi-omics approaches, integrating single-cell transcriptomics, chromatin accessibility profiling, and spatial transcriptomics to gain a more comprehensive view of cellular states. Future studies should continue to expand on this by incorporating proteomics, metabolomics, and functional gene knockout and knockdown experiments to further bridge the gap between transcriptomic signatures and their phenotypic consequences [51]. Nevertheless, several studies have already demonstrated the biological relevance of SPP1 and shed light on its therapeutic potential.

## 7. Therapeutic Potential

SPP1 is a secreted glycoprotein that plays a critical role in fibrosis by regulating extracellular matrix remodeling, fibroblast activation, and immune cell recruitment [7,8]. Although macrophages are a significant source of SPP1, other cell types, including fibroblasts and epithelial cells, also contribute to its secretion [9]. Given its broad functional role, the therapeutic targeting of SPP1 using antibodies may exert antifibrotic effects through several potential mechanisms.

One primary mechanism is the neutralization of SPP1 signaling, where antibody binding prevents SPP1 from interacting with its receptors, including integrins (e.g., α5β1, αvβ3, α4β1) and CD44, thereby attenuating pro-fibrotic signaling cascades [10,11]. This disruption can reduce fibroblast activation and collagen deposition, both of which are key features of fibrosis progression. Additionally, antibody–SPP1 complexes may be subject to enhanced clearance by macrophages, particularly through Fc receptor-mediated phagocytosis, leading to a reduction in SPP1 bioavailability and downstream fibrotic effects [12].

Moreover, SPP1 has been implicated in macrophage polarization toward a pro-fibrotic phenotype, and its inhibition may lead to a shift towards a less pro-fibrotic macrophage state [13]. This modulation of macrophage polarization could indirectly attenuate fibrosis by reducing TGFβ, NF-κB, and TNF-signaling [5]. Furthermore, given that SPP1 serves as a mediator of intercellular communication between macrophages, fibroblasts, and immune cells, its neutralization by antibodies may disrupt cell–cell communication, further limiting fibroblast proliferation, migration, activation, and extracellular matrix deposition [5,14].

Former studies utilizing a *Spp1* knock-down or a neutralization of SPP1 have highlighted its significant role in the development and progression of fibrosis across various organs. It has been implicated in promoting fibrotic processes through multiple mechanisms. In the kidney, SPP1 facilitates macrophage recruitment to postischemic tissue, inhibits apoptosis, and stimulates the development of renal fibrosis following acute ischemic injury [52]. Similarly, in the liver, SPP1 has been shown to delay the resolution of fibrosis, suggesting its potential as a therapeutic target in liver diseases [53]. In the context of non-alcoholic steatohepatitis (NASH), neutralizing antibodies against SPP1 have demonstrated efficacy in attenuating disease progression in mouse models [54]. Furthermore, in vitro studies have shown that *Spp1* knock-down and SPP1 neutralization can abrogate leptin-mediated fibrogenesis [55]. Notably, recent research in pulmonary fibrosis has revealed that inhibition of SPP1 can attenuate lung fibrosis, highlighting a potential therapeutic avenue for treating this condition [56]. The pro-fibrotic effects of SPP1 extend to the heart, where it plays a role in cardiac fibrosis and remodeling associated with angiotensin II-induced cardiac hypertrophy [57].

The latest advances facilitated by next-generation sequencing have refined the understanding of *SPP1*’s role in fibrosis, narrowing its contribution down from a previously rather broad and unspecific involvement to its specific expression in mononuclear cells, particularly macrophages, as key mediators of fibrotic processes.

Momin et al. built upon this knowledge in their study by developing an antibody–siRNA conjugate (ARC) specifically designed to target and silence *Spp1* in TREM2+ cardiac macrophages as a novel immunotherapy for the treatment of atrial fibrillation (AFib) [58]. Given that cardiac fibrosis significantly contributes to major arrhythmias by impairing electrophysiological processes, and TREM2+ SPP1+ macrophages are established promoters of fibrosis [18], the therapy employs an ARC specifically engineered to diminish *Spp1* in TREM2+ cardiac macrophages. In preclinical studies, the ARC achieved up to a 90% reduction in *Spp1* expression in TREM2+ macrophages within both murine and human cardiac tissues. In a mouse model of AFib (HOMER), ARC treatment significantly reduced atrial fibrosis, decreased AFib inducibility from 82% to 33%, and shortened total AFib episode duration by four-fold. SnRNA-seq further revealed a reduction in pro-fibrotic fibroblasts and their associated collagen expression following ARC administration. Whether this therapeutic approach can also be applied to left ventricular fibrosis remains to be determined by further studies; however, the strategy appears highly promising.

Although these findings have not been officially peer-reviewed and published at the current time of writing, they nonetheless highlight the therapeutic potential of targeting *SPP1* in fibrotic disorders [58].

## 8. Conclusions

In conclusion, this review underscores the pivotal role of SPP1+ macrophages in the pathogenesis of multiorgan and especially cardiac fibrosis. Chronic heart failure, characterized by high morbidity and mortality rates, presents a significant challenge due to its limited therapeutic options. The emerging focus on cardiac fibrosis as a therapeutic target highlights the need for novel approaches to mitigate adverse cardiac remodeling, which traditional anti-inflammatory strategies have failed to address effectively.

SPP1+ macrophages have been identified as key regulators in the fibrotic process, promoting ECM remodeling and interacting closely with stromal cells and myofibroblasts. SPP1+ macrophages appear to be monocyte-derived macrophages that proliferate within scar tissue and exhibit pro-fibrotic properties. While there is a clear overlap with TREM2+ macrophages, not all SPP1+ macrophages seem to express TREM2. Therefore, we believe it is reasonable to consider SPP1+ macrophages as a distinct subset. Their involvement in various forms of fibrosis across multiple organs, including the heart, liver, lungs, and kidneys, points to their critical role in tissue repair and fibrosis progression. The relevance across organs suggests a conserved mechanism of fibrosis and highlights the tremendous potential of this subpopulation for the treatment of fibrosis.

Future therapeutic strategies could benefit from directly silencing *SPP1* in macrophages, modulating or intercepting specific signaling pathways, or targeting distinct macrophage populations to regulate their activity and mitigate fibrosis. The identification of pathways and molecules, such as CXCL4 and GM-CSF, that regulate the differentiation and function of SPP1+ macrophages highlights additional promising therapeutic targets.

By advancing our understanding of these cellular processes, we can pave the way for the development of more effective treatments for cardiac fibrosis and chronic heart failure, ultimately improving patient outcomes and reducing the burden of these debilitating conditions.

## Figures and Tables

**Figure 1 cells-14-00345-f001:**
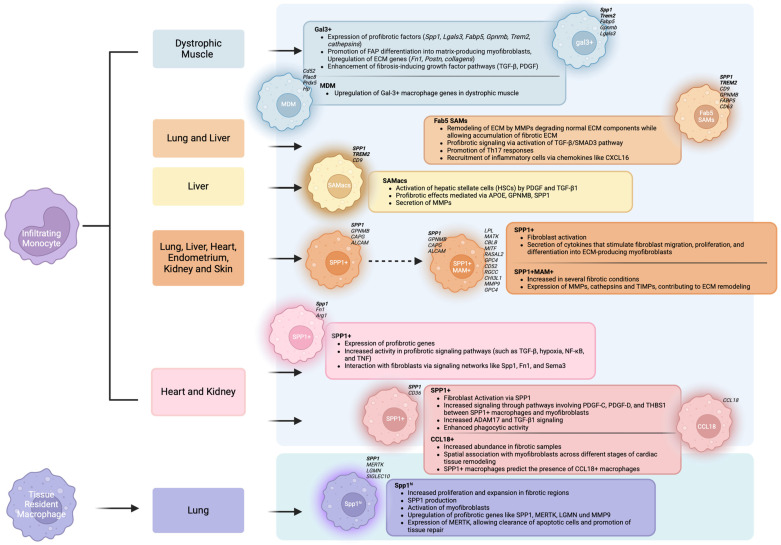
Graphical abstract: Current knowledge of pro-fibrotic SPP1+ macrophage subtypes from single-cell transcriptomics across organs (Gal3+ and MDM [43], Fab5 SAMs [40], SAMacs [41], SPP1+ MAM−/+ [44], SPP1+ [17,18], SPP1^hi^ [45]).

**Figure 2 cells-14-00345-f002:**
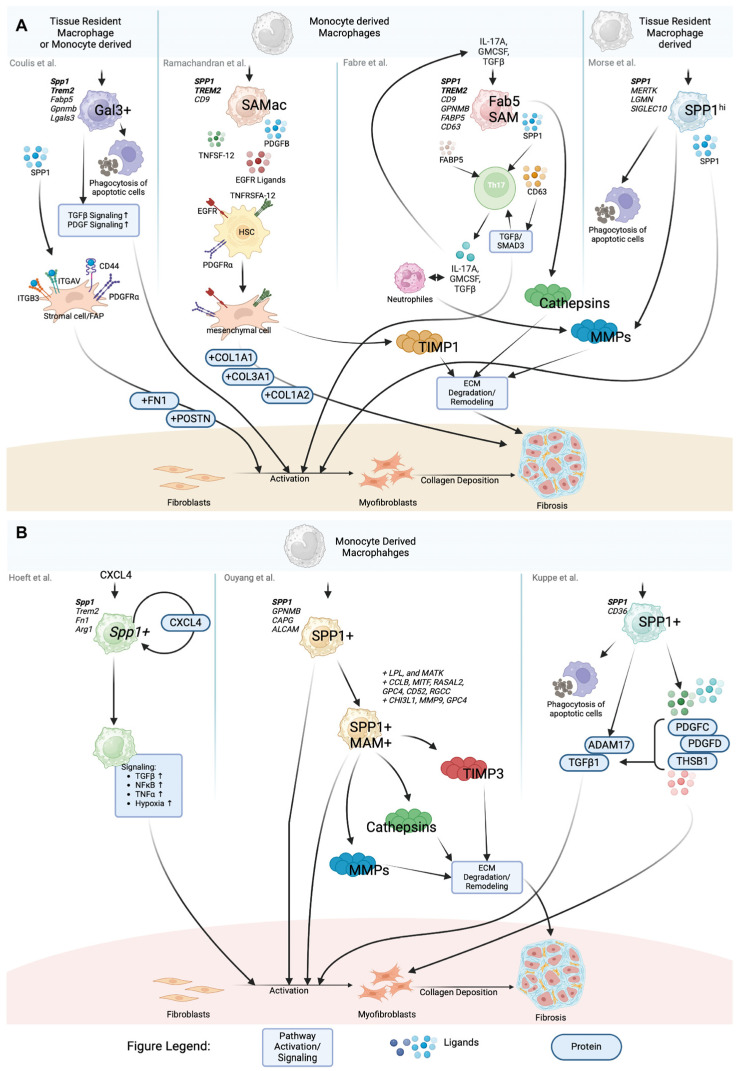
SPP1+ macrophages in fibrosis. (**A**) Schematic representation of SPP1+ macrophages and their pro-fibrotic effects as described in organs other than the heart (e.g., dystrophic muscle, lung, liver). (**B**) Schematic depiction of SPP1+ macrophages specifically identified in cardiac fibrosis and their associated pro-fibrotic mechanisms.

**Table 1 cells-14-00345-t001:** Summary of pro-fibrotic macrophage subtypes: authors, marker genes, localization, and functions.

Authors	Models Used		Methods Used	Subtype	Signature	Species	Marker Genes
**Coulis et al.** 10.1126/sciadv.add9984	1. C57BL/10ScSn-Dmdmdx/J mice (referred to as B10.mdx or mdx mice), a model for Duchenne muscular dystrophy		scRNAseq, bulk RNAseq, spatial transcriptomics	Gal-3+ macrophages	Gal3+	mouse	***Spp1***, ***Trem2***, *Fabp5*, *Gpnmb*, *Lgals3*
				Monocyte-derived macrophages (MDMs)	MDMs	mouse	*Cd52*, *Plac8*, *Prdx5*, *Hp*
**Ramachandran et al.** 10.1038/s41586-019-1631-3	C57BL/6JCrl mice treated with carbon tetrachloride (CCl4), a model of liver fibrosis, human liver tissues for comparison and cross-species analysis		scRNA-seq and spatial mapping	scar-associated macrophages (SAMacs)	SAMac	mouse	***Spp1***, ***Trem2***, *Cd9*, *Lgals3*
						human	***SPP1***, ***TREM2***, *CD9*
**Fabre et al.** 10.1126/sciimmunol.add8945	human liver and lung single cell RNA sequencing datasets, C57BL/6JCrl mice treated with carbon tetrachloride (CCl4), a model of liver fibrosis, bleomycin lung injury a model of pulmonary fibrosis, high-fat diet a model of NASH		scRNA-seq	“Fab5” or five marker-positive scar-associated macrophages (SAM)	Fab5 SAMs	mouse	***Spp1***, ***Trem2***, *Cd9*, *Fabp5*, *Gpnmb*, *Cd63*
						human	***SPP1***, ***TREM2***, *CD9*, *GPNMB*, *FABP5*, *CD63*
**Morse et al.** 10.1183/13993003.02441-2018	human tissue samples of healthy and IPF Lungs		scRNA-seq	SPP1-hi Macrophages	SPP1-hi	human	***SPP1***, *MERTK*, *LGMN*, *SIGLEC10*
**Ouyang et al.** 10.7554/eLife.85530	multiple human tissues: liver: cirrhosis and nonalcoholic steatohepatitis (NASH), lung: idiopathic pulmonary fibrosis (IPF) and systemic sclerosis (SSc), heart: ischemic cardiomyopathy (ICM) and dilated cardiomyopathy (DCM), skin: keloid scarring and systemic sclerosis (SSc), endometrium: endometriosis, kidney: chronic kidney disease (CKD) and acute kidney injury (AKI)		integration of 15 scRNA-seq datasets from multiple human tissues	SPP1+ macrophages	SPP1+	human	***SPP1***, *GPNMB*, *CAPG*, *ALCAM*
				SPP1+ MAM+ macrophages	SPP1+MAM+	human	***SPP1***, *GPNMB*, *CAPG*, *ALCAM*, *+LPL*, *and MATK+CCLB*, *MITF*, *RASAL2*, *GPC4*, *CD52*, *RGCC +CHI3L1*, *MMP9*, *GPC4*
**Hoeft et al.** 10.1016/j.celrep.2023.112131	analysis of human kidney and heart tissue samples from patients with chronic kidney disease and heart failure, myocardial infarction model in wild-type and Cxcl4^−/−^ mice, renal unilateral ischemia-reperfusion injury model in wild-type and Cxcl4^−/−^ mice, bone marrow transplantation to generate hematopoietic Cxcl4 knockout mice		scRNA-seq, snRNA-seq	*Spp1+* macrophages	*Spp1+*	mouse	***Spp1***, *Trem2*, *Fn1*, *Arg1*
				SPP1+ macrophages	SPP1+	human	***SPP1***, ***TREM2***, *CD9*, *FN1*, *APOE*
**Kuppe et al.** 10.1038/s41586-022-05060-x	Human heart tissue, myocardial infarction model in C57Bl/6J Pdgfrb-creER;tdTomato mice		snRNA-seq, snATAC-seq, spatial transcriptomics	SPP1+ macrophages	SPP1+	human	***SPP1***, *CD36*
				CCL18+ macrophages	CCL18+	human	*CCL18*
**Authors**	**Signature**	**Localization**	**Pro-Fibrotic Effects**				
**Coulis et al.** 10.1126/sciadv.add9984	Gal3+	dystrophic muscle	1. Expression of pro-fibrotic factors (*Spp1*, *Lgals3*, *Fabp5*, *Gpnmb*, *Trem2*, *cathepsins*) by Gal-3+ macrophages. 2. Colocalization of Gal-3+ macrophages with PDGFRα+ stromal cells in dystrophic lesions. 3. Spp1-mediated signaling between Gal-3+ macrophages and stromal cells, primarily through SPP1–CD44 interaction. 4. Potential promotion of FAP differentiation into matrix-producing myofibroblasts by Gal-3+ macrophages. 5. Chronic activation of Gal-3+ macrophages in muscular dystrophy, leading to increased collagen deposition. 6. Upregulation of ECM genes (*Fn1*, *Postn*, *collagens*) and fibrosis-inducing growth factor pathways (TGFβ, PDGF) in Gal-3-enriched areas.				
	MDMs	dystrophic muscle	1. MDMs upregulate gal-3+ macrophage genes when exposed to the dystrophic muscle environment.				
**Ramachandran et al.** 10.1038/s41586-019-1631-3	SAMac	collagen-rich scar regions of the liver, originate from circulating monocytes	1. Expression of profibrogenic genes including *SPP1*, *LGALS3*, *CCL2*, *CXCL8*, *PDGFB,* and *VEGFA*. 2. Conditioned medium from SAMacs promotes fibrillar collagen expression in primary human hepatic stellate cells (HSCs). 3. Localization in collagen-positive scar regions in cirrhotic livers. 4. Expression of EGFR ligands that are known to regulate mesenchymal cell activation. 5. Expression of mesenchymal cell mitogens TNFSF12 and PDGFB, which signal to cognate receptors TNFRSF12A and PDGFRA on scar-associated mesenchymal (SAMes) cells. 6. Induction of proliferation of human HSCs by TNFSF12 and PDGF-BB from SAMacs. 7. Conditioned medium derived from SAMacs enhances the proliferation of primary human HSCs in ex vivo cultures.				
**Fabre et al.** 10.1126/sciimmunol.add8945	Fab5 SAMs	expand in number during fibrosis progression in both liver and lung; localize to the edges of fibrotic scars, at the interface between pathogenic fibrillar ECM and normal ECM	1. ECM remodeling by degradation of regular ECM components and accumulation fibrillar collagen I. 2. Stimulation of fibroblasts and myofibroblast differentiation and enhancement of collagen I deposition. 3. Amplification of pro-fibrotic signaling by activation of TGF-β/SMAD3 pathway. 4. Promotion of Th17 responses. 5. Recruitment of inflammatory cells due to expression of chemokines like CXCL16. 6. Sustaining fibrosis-promoting inflammatory milieu by inducing Th17 cells. 7. Increase in MMP activity targeting primarily normal ECM substrates.				
**Morse et al.** 10.1183/13993003.02441-2018	SPP1-hi	present in normal lungs but greatly expanded in IPF lungs; dramatically increased in number and SPP1 expression in fibrotic IPF lower lobes; associated with fibroblastic foci in IPF lungs; show increased proliferation in IPF, especially in fibrotic lower lobes	1. Increased proliferation and expansion in fibrotic regions. 2. High expression and deposition of osteopontin (SPP1) in the extracellular matrix, especially around fibroblastic foci. 3. Upregulation of pro-fibrotic genes like SPP1, MERTK, LGMN, und MMP9. 4. Expression of MERTK, allowing clearance of apoptotic cells and promotion of tissue repair. 5. Potential direct activation of myofibroblasts, suggested by graphical modeling. 6. Upregulation of genes involved in extracellular matrix organization. 7. Possible phagocytosis of damaged alveolar epithelial cells through MERTK expression. 8. Production of MMP9, which is involved in matrix remodeling.				
**Ouyang et al.** 10.7554/eLife.85530	SPP1+	healthy and fibrotic tissue across all six organs examined, proportions tended to be increased in fibrotic conditions compared to healthy controls; tissues like liver, endometrium and kidney had relatively higher levels of SPP1+ MAM− macrophages compared to healthy lung, heart, and skin; FCN1+ monocytes enter the tissue and begin differentiating via transitional macrophages into SPP1+ MAM− macrophages	1. SPP1+ MAM− macrophages are enriched for secreted factors and inflammatory/immune pathways, which likely promote fibroblast activation. 2. These macrophages secrete cytokines that stimulate fibroblast migration, proliferation, and differentiation into ECM-producing myofibroblasts. 3. They contribute to collagen I-associated fibrosis through their secretory activity. 4. SPP1+ MAM− macrophages are positively associated with aging in the lung, promoting chronic low-grade inflammation (inflammaging), which can trigger early fibrogenesis.				
	SPP1+ MAM+	both healthy and fibrotic tissue across all six organs examined; proportions were significantly increased in fibrotic conditions compared to healthy controls in several tissues, including IPF lung, SSc lung, keloid skin, and AKI kidney; SPP1+MAM- macrophages polarize to SPP1+ MAM+ macrophages	1. Expression of matrix metalloproteinases (MMP7, MMP9, MMP19), their inhibitors (TIMP3), and cathepsin genes (CTSK) contributing to ECM remodeling. 2. SPP1+ MAM+ macrophages show enrichment for metabolic processes, which may support their pro-fibrotic functions. 3. They are associated with osteoclast development pathways, suggesting similarities to bone-remodeling cells. 4. SPP1+ MAM+ represent an advanced polarization state arising from SPP1+ macrophages in fibrotic conditions. 5. Their proportions are significantly increased in several fibrotic conditions compared to healthy controls.				
**Hoeft et al.** 10.1016/j.celrep.2023.112131	*Spp1+*	the paper does not provide specific information about the localization of the pro-fibrotic Spp1+ macrophages within organs; they expand after organ injury in both the heart and kidney	1. Expression of pro-fibrotic genes: *Spp1*, *Fn1,* and *Arg1.* 2. Increased activity of pro-fibrotic signaling pathways: TGF-β, hypoxia, NF-κB, and TNF signaling. 3. High activity of transcription factors associated with fibrosis: Hif1a, Myc, and Spi1. 4. SPP1+ macrophages show the highest ECM regulator scores among immune cells. 5. Interaction with and activation of fibroblasts through signaling networks: *Spp1*, *Fn1*, and *Sema3.* 6. Persistence and expansion of SPP1+ macrophages during the remodeling phase, day 3–7 after MI.				
	SPP1+						
**Kuppe et al.** 10.1038/s41586-022-05060-x	SPP1+	increased in abundance in ischemic samples compared to other sample types; spatially associated with myofibroblasts, particularly in areas of cardiac remodeling after myocardial infarction; enriched in the inflammatory cell-type niche (niche 5) identified in the spatial transcriptomics analysis; validated by RNA in situ hybridization to be spatially interacting with myofibroblasts in human cardiac tissues following myocardial infarction	1. SPP1+ macrophages show increased abundance in ischemic samples. 2. SPP1+ macrophages have a spatial association with myofibroblasts across different stages of cardiac tissue remodeling. 3. There is increased PDGF-C, PDGF-D, and THBS1 signaling between SPP1+ macrophages and myofibroblasts in ischemic versus myogenic samples. 4. Increased ADAM17 and TGFB1 signaling between SPP1+ macrophages and fibroblasts in fibrotic versus myogenic samples. 5. SPP1+ macrophages show increased phagocytic activity, as evidenced by multiple intracellular vacuoles.				
	CCL18+	increased in fibrotic samples	1. Increased abundance in fibrotic samples. 2. Spatial association with myofibroblasts across different stages of cardiac tissue remodeling. 3. SPP1+ macrophages predict the presence of CCL18+ macrophages				

## Data Availability

The data supporting this narrative review are openly available in the published literature and can be accessed through the references cited throughout the article. No new data were generated or analyzed in this study, as it is based on existing published research and literature.

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
