# Peer review of "Pro-Fibrotic Macrophage Subtypes: SPP1+ Macrophages as a Key Player and Therapeutic Target in Cardiac Fibrosis?"

_cells, 2025, doi:10.3390/cells14050345_

Round 1

Reviewer 1 Report

Comments and Suggestions for Authors

General comments

In this manuscript, the authors reviewed recent advances on the role of SPP1+ macrophages in cardiac fibrosis, with a particular focus on their involvement in fibrosis progression and their therapeutic potential. They highlighted findings from single-cell transcriptomics identifying SPP1+ macrophages as key mediators of fibrosis. Furthermore, the review explored preclinical therapeutic strategies targeting these macrophages, including silencing Spp1 with antibody-siRNA conjugates and modulating differentiation pathways. The authors underscored the potential of targeting SPP1+ macrophages as a promising approach for developing therapies for heart failure. The review addresses a highly relevant and timely topic. However, as it is presented in the form of a minireview, some sections are difficult to follow due to the lack of details.

Specific comments

1.  To enhance clarity and accessibility, the authors are encouraged to include a figure illustrating the function of SPP1+ macrophages under both physiological and pathological conditions. This would allow for a more comprehensive understanding of these cells and their roles.

2.   The text and layout of Table 1 are problematic due to the extremely small font size, which renders it nearly illegible. Describing the table using two pages may improve readability.

3.  For Figure 1, a more schematic representation describing the function and markers of each macrophage subtype would be beneficial. Instead of using sentences, summarizing the functions as concise highlights would make the information easier to understand. The markers should also be clearly delineated.

4. While the review provides valuable insights into SPP1+ macrophages, it does not address many other macrophage subtypes. A comprehensive table summarizing all macrophage subtypes, including their key characteristics, would give readers a broader perspective and help position SPP1+ macrophages within the context of these newly described types. The authors may find it helpful to reference a recent and comprehensive article review on macrophage subtypes (PMID: 38816371) to further enrich their description of the cells.

Reviewer 2 Report

Comments and Suggestions for Authors

In this review article, Uhlig and colleagues discuss the role of subsets of macrophage populations in fibrotic diseases in particular SPP1+ macrophages and their implications in cardiac fibrosis. Overall, it is a reasonable discussion centering on recent emphasis on delineating particular cellular populations that play a role in diseases through recent technologies in particular transcriptomics analyses. I have the following comments.

Major:

1) Title suggests the focus on "cardiac fibrosis" but the direction of the subsection suggest generalization of fibrosis. There has not been much description of the roles of cardiac fibrosis in particular or the implications it has for organ function. However, the authors do briefly touch on atrial fibrillation, myocardial infarction, heart failure.

2) Figure 1 - Standardize the first letter capitalization or not - e.g. "dystrophic muscle", "Lung and Liver" etc - should the next organ after the comma be first capitalized or not?

"gal3+" vs. "SPP1+" vs. "Fab5"- standardization. Suggest to follow HUGO/MGI nomenclature (depending on species in reference throughout the text) when refering to genes and/or protein - emphasis on italics/capitalizations differences between species and protein vs. genes. Also, "GAL-3" or "gal3" etc...

What evidence is there that hypoxia, NFkb, TNF are part of the profibrotic signaling pathways i.e. they appear inflammatory insults or are they a stimuli for promoting fibrosis? Signaling pathways often refer to downstream mediators e.g. SMADs, STATs, ERK, AKT, etc... rather than upstream cytokines or insults.

The figure appear to refer to published literature with regards to specific organs rather than general "current knowledge". Please cite the references in the figure if that's the case e.g. [1], [2], etc

3) Page 3 Fibrosis - The authors discuss about the role of reparative fibrosis but not replacement fibrosis due to cellular apoptosis/necrosis e.g. MI-scarring - this is often considered irreversible

4) Table 1 - Text are too small to read.

5) Page 7 Line 177 - i'm unfamiliar with the term "type 3 inflammation" - you referring to type III hypersensitivity (immune complex-mediated inflammation)?

6) Page 7 Line 186-206 and beyond - it is confusing when authors refer to gene expression and species of model...e.g. SPP1 (protein) or Spp1 (italics, gene in mouse) especially since reference is made to scRNAseq which measures gene expression, but if protein validation were made e.g. through flow analyses for high vs. low protein expression population.... especially since osteopontin is a secreted glycoprotein.

7) It may be beneficial to generate a second figure that compares the earlier conception of M1/M2 macrophages and their markers and implications to disease vs. the current specialized subsets of macrophages (LAM, SAM, MAMs etc) how do they differ in marker expressions and in what context/organs/models are they expressed.

8) Page 9 Therapeutic Potential - Given that the SPP1 is a secreted glycoprotein, and not necessarily only secreted by macrophages, can the authors speculate on mechanisms of effect? e.g. how do antibodies that bind SPP1 affect fibrosis... do they directly get cleared by other macs?

With regards to scRNAseq/transcriptomics, i personally feel that there should be a limitation discussion which is often overlooked: (1) RNA is not = protein, these should be validated, (2) the cell population characterization largely depends on depth of sequencing and (3) tissue isolation strategy, etc etc. especially with regards to rare cell populations...

10) Page 10 Line 327 - nothing wrong with being a preprint - these are still citable. Perhaps the correct implication is to say these findings have not been officially peer-reviewed and published at the current time of writing.

11) References - 18/43 refs (~42%) >7 years old; oldest from 1998. Please check if there may be updated references to replace. Also check if refs are outdated reviews? original articles - these are fine if data is directly cited to support your discussion; but if reviews - check that data is still relevant or can be replaced with more up-to-date discussions.

Minor:

1) Page 3 SPP1 - "TH-1" check appropriate nomenclature. Th-1 or Th1 or TH-1.

2) Acknowledgement - the weblink is not finalized - that is OK; only needed for final proofs stage. 

3) Abbreviations - standardize abbreviation descriptions - some small letters some first letter capitalized some all words captalized...

4) Abbreviation Lgals3 (gene) and Gal-3 (protein) - not quite clear in the text but reference to either crops up intermittently.

Reviewer 3 Report

Comments and Suggestions for Authors

The review is well conducted and comprehensively analyzes the currently available studies on the role of SPP1 macrophages in the pathogenesis of multiorgan and cardiac fibrosis; furthermore, the authors review and discuss the potential future clinical implications arising from the advancement of the knowledge of these cellular processes and the development of therapeutic strategies specifically aimed at targeting and silencing SPP1 in macrophages, which could ultimately lead to the treatment and containment of fibrotic disorders and their negative consequences on the function of the affected organs.

The authors make a commendable effort in synthesizing the current knowledge on profibrotic macrophage subtypes in a graphical abstract and a summary table that make the very technical topic easier to understand even for non-specialist readers.

The only suggestion I have is to reread the English text to correct some small spelling errors.

Round 2

Reviewer 1 Report

Comments and Suggestions for Authors

The authors responded to all my concerns

Comments on the Quality of English Language

It is good.

Reviewer 2 Report

Comments and Suggestions for Authors

I am satisfied with the authors' responses and amended manuscript. I have no further suggestions.